# Impact on Body Composition and Physical Fitness of an Exercise Program Based on Immersive Virtual Reality: A Case Report

**DOI:** 10.3390/jfmk10010056

**Published:** 2025-02-04

**Authors:** Andrés Godoy-Cumillaf, Paola Fuentes-Merino, Frano Giakoni-Ramírez, Marcelo Maldonado-Sandoval, José Bruneau-Chávez, Eugenio Merellano-Navarro

**Affiliations:** 1Grupo de Investigación en Educación Física, Salud y Calidad de Vida, Facultad de Educación, Universidad Autónoma de Chile, Temuco 4780000, Chile; andres.godoy@uautonoma.cl (A.G.-C.); paola.fuentes@uautonoma.cl (P.F.-M.); marcelo.maldonado@cloud.uautonoma.cl (M.M.-S.); 2Facultad de Educación y Ciencias del Deporte, Instituto del Deporte y Bienestar, Universidad Andres Bello, Santiago 7550000, Chile; frano.giakoni@unab.cl; 3Departamento de Educación Física, Deportes y Recreación, Universidad de la Frontera, Temuco 4811230, Chile; jose.bruneau@ufrontera.cl; 4Department of Physical Activity Sciences, Faculty of Education Sciences, Universidad Católica del Maule, Talca 3530000, Chile

**Keywords:** fat, muscle, cardiorespiratory fitness, speed-agility, strength

## Abstract

**Background/Objectives:** The practice of physical activity contributes to obtaining adequate values of body composition and physical fitness, which is beneficial for people’s health. However, a large part of the adult population does not comply with the recommendations for physical activity, due to factors such as lack of time and nearby sports venues. Immersive virtual reality is a tool that allows individuals to immerse themselves in a simulated world and perceive visual, auditory, and tactile sensations. Its use in physical activity interventions favors exercise in situations that, due to time and space, could be limited in real life. The objective of this case report is to measure the impact on body composition and physical fitness of an exercise program executed through immersive virtual reality. **Methods**: The design is a case study with a quantitative approach developed through a physical activity intervention with immersive virtual reality in which body composition was evaluated considering fat and muscle components, and physical fitness considering cardiorespiratory fitness, speed-agility, and hand grip strength, through pre- and post-testing. The physical exercise program based on immersive virtual reality lasted 8 weeks. The subject of the study was a 24-year-old man, a second-year student of Pedagogy in Physical Education at a Chilean university, with no previous experience in this virtual tool. **Results**: The results indicate that for body composition the study subject decreased the fat component and slightly improved the musculature, while for physical fitness cardiorespiratory fitness and speed-agility improved, but manual grip strength decreased. **Conclusions**: It is concluded that the training developed through immersive virtual reality proves to be a tool that can promote improvements in body composition and physical fitness; it is necessary to carry out more research to validate the potential of this instrument as a means of contributing to the health of the population.

## 1. Introduction

Physical fitness (FP) integrates various bodily functions involved in daily physical activity or exercise [1]. Maintaining adequate PF levels is crucial for health, as it correlates with better cardiovascular, skeletal, and mental health [2,3]. FP is therefore an essential component of a healthy lifestyle [4]. In addition, body composition quantifies and examines relationships and changes in components of the human body [5]. Understanding body composition is vital for assessing the impact of factors such as growth, diet, disease, and physical exercise on the body [6]. Research indicates that physical activity interventions aimed at improving FP result in better body composition, specifically fat reduction and increased muscle mass [7,8]. These changes benefit health by reducing the risk of metabolic diseases and insulin resistance associated with excess fat, particularly in the waist and abdomen [9], while adequate muscle mass improves metabolic functions in the liver, pancreas, and immune system [10].

Despite well-documented health benefits, more than a quarter of adults worldwide remain physically inactive [11]. Factors that contribute to this inactivity include the perception of exercise as boring or difficult, long working hours, lack of motivation, insufficient family and social support, and limited access to sports facilities [12,13]. This problem is particularly pronounced among young adults (defined as 18–25 years old), whose lifestyles are shaped by access to technology, digital nativity, and economic support through state grants and scholarships. In addition, the COVID-19 pandemic’s shift to virtual education has affected their learning and social interactions, exacerbating health issues that require early intervention for a healthier adulthood [14,15].

In terms of health indicators, young adults belonging to Generation Z tend to have higher rates of mental health diagnoses, shorter sleep quality duration, and other conditions related to poor health habits, such as high rates of physical inactivity, increased sedentary behavior, or less time for sports [16]. On the other hand, this generation in its adolescence, according to data from the latest JUNAEB Nutritional Map, had a prevalence of overweight/obesity of more than 50% [17].

An effective tool to increase physical activity and improve health parameters in this population could be immersive virtual reality (IVR). IVR is a technique in which individuals, through a head-mounted screen, receive visual, auditory, and tactile sensations of a simulated world [18]. Through an avatar, participants can perform movements that align with their body’s actions from a first-person perspective. This immersive experience has participants adjust their actual actions based on the avatar’s movements, driven by the illusion of possessing the avatar’s body [18]. This illusion generates measurable motor and physiological responses in the real body, allowing individuals to participate in situations that would otherwise be impossible due to time or space constraints [19].

As a result of the benefits that IVR brings, fields related to public health have integrated IVR, since it presents the possibility of strengthening repetitive tasks and increasing graphic and acoustic feedback [20]. This has been reflected in several studies that describe the effects of programs implemented through IVR in the rehabilitation of patients with cardiovascular [21], Parkinson’s [22], and obstructive pulmonary diseases [23], among others. On the other hand, several studies have shown that through the use of IVR, it is possible to replicate and even improve the results of physical capacity variables that are related to health, in relation to those performed with traditional exercise (e.g., conventional training, treadmill walking, cycling) [24,25,26,27,28], also generating greater enjoyment [29]. This is in addition to the fact that evidence places it as a tool that promotes healthy behaviors and the practice of physical activity, increasing adherence to exercise in the long term [19]. A recent exploratory review established that its use stimulates moderate and vigorous intensity levels, which could contribute to meeting physical activity recommendations [30], while other studies conclude that the practice of physical exercise through IVR causes moderate- and vigorous-intensity physical activity [31,32], which are the levels that provide health benefits. On the other hand, there is information that indicates that performing physical activity through IVR generates a high level of enjoyment in participants, it is perceived as a safe environment [33], it produces greater calorie burning, greater concentration, and the feeling of being able to continue exercising for longer [34], and it allows monitoring and obtaining immediate feedback in terms of energy expended, intensity achieved, and power generated, among others.

There is evidence that the use of immersive virtual reality can bring post-execution balance problems, headaches, and nausea as side effects [35,36]; however, taking into account that the practice of physical exercise in the young adult population is low and that IVR has been shown to obtain similar or better results than the execution of traditional physical activity, in addition to presenting greater enjoyment and adherence as well as providing a safe environment, it could be an alternative way to overcome the limitations that prevent the practice of physical exercise in adults. In addition, due to the characteristics of the study subject, IVR could provide relaxing virtual environments that help reduce stress and anxiety through the practice of physical activity, favoring their university life. For all of the above, this case report aimed to measure the impact on body composition and physical fitness of an exercise program executed through immersive virtual reality. The information that may be useful for this emerging line of research, especially in the design of physical activity programs through IVR in young adults who are inserted into university life.

## 2. Materials and Methods

This case report was approved by the Scientific Ethics Committee of the Universidad Autónoma de Chile (CEC-N°42-22) and followed CARE’s guidelines for case reports [37]. Prior to the start of the research, the study subject gave written consent to participate.

### 2.1. Subject

The subject of the study was a 24-year-old man, a second-year student of Pedagogy in Physical Education at a Chilean university. He had previous experience with the practice of physical activity (he constantly practiced physical activity through the practice of sports for recreation, through games, or through traditional training (e.g., jogging, cycling). He did not follow a physical exercise routine developed by a specialist, nor did he practice competitive sports. At the time of the study, he did not have musculoskeletal problems or chronic diseases that prevented him from performing the physical exercise program. As for his experience with immersive virtual reality, he had never used it. He was asked not to perform any other type of physical activity throughout the intervention.

### 2.2. Study Variables

Before the physical exercise program, and after its completion, the physical fitness of the participant was evaluated under the same conditions in both measurements. Specifically, the speed-agility component was evaluated through the 4 × 10 test, where two lines separated from each other by 10 m are demarcated on the floor. On the starting line is located a sponge (B), and on the other line are located two sponges (A and C). When instructed, the assessee ran as fast as possible to the other line, took one of the sponges (A), and returned to the starting line; when he arrived, he changed sponges (he left sponge A and took sponge B), ran again to the other line, changing sponges (leave sponge B and take sponge C), and returned to the starting line. The time was measured from when he first left the starting line until he completely passed the starting line after taking the last sponge (C). The best of two attempts was recorded.

The upper-body muscle strength component was assessed through the hand-grip strength test with a digital dynamometer (takei 5401). The person being evaluated took the dynamometer with one hand and squeezed as hard as possible, continuously and for two seconds, without the instrument touching his body. An average of two attempts were recorded for each hand.

Cardiorespiratory fitness was assessed through the Course Navette test, for which two separate marks are drawn at 20 m. The subject had to run between the marks, adapting his running speed, to reach the marks at the same time as a sound stimulus was emitted. It started with a speed of 8.5 km/h and increased by 0.5 km/h every minute. The test ended when on two occasions the subject did not reach the marks along with the sound stimulus. The last minute reached was recorded. The maximum oxygen consumption was estimated through the formula proposed by Leger et al. (1988) [38]. All the instruments used to measure the physical fitness components are widely used in research of this nature and have adequate levels of validity and reliability [38,39,40]. For body composition, fat and muscle components were evaluated, through the measurement of bone diameters, body perimeters, and skin folds, according to the protocol described by the International Society for the Advancement of Kineanthropometry [41]. The evaluator was the same in both measurements and had ISAK level 3 certification and a technical measurement error of 0.7%.

### 2.3. Physical Exercise Program

The program was based on training inspired by martial arts, where touch controls simulate boxing gloves and combinations of punches to spherical objects are made, in addition to avoiding being hit by blocks by performing lateral movements that are mostly squats. Three levels were used (beginner, intermediate, and advanced), which became more complex as the speed of execution increased. The entire exercise program was performed using the FitXR app (version 3.7.88, developed by FITXR LIMITED, London, UK). The FitXR app was chosen because it has been proven to be easy to use for people with no prior RVI experience [42]. In addition, it has already been used in a population with characteristics similar to the study subject [32,43], and there is evidence that shows that it causes moderate and vigorous physical activity intensities [32], which are recommended for their benefits in physical fitness and body composition [44].

### 2.4. Sessions

The physical exercise program was carried out three times a week for 8 weeks, giving a total of 24 sessions (in Appendix A, you can see the number of minutes per intensity of physical activity in each of the sessions developed). Throughout the intervention, a Physical Education teacher, a specialist in virtual reality, was present at all times. Before starting the intervention, a familiarization session was held, where the team presented the implements and characteristics of the devices. Each session was divided into three parts, the first was the beginning where a warm-up was carried out, consisting of oxygenation, joint mobility, and stretching, lasting 15 min, at an intensity of 60% to 70% of the maximum heart rate (MHR). The second part was where the physical exercise program was executed in its three levels, beginner, intermediate, and advanced, with a duration of 8, 9, and 8 min, respectively. The intensity of the work was between 80% and 95% of the MHR. Between each level there was a 5 min break where the participant rested, hydrated, and dried their sweat. The third part of the session consisted of a cool-down that lasted 10 min at an intensity of 50% to 60% of the MHR, followed by 10 min of stretching. All three parts of the session were executed via IVR. Each session lasted a total of 1 h.

The intensity of the exercise was determined by means of a maximal cardiorespiratory effort test, which was executed on a Monark LC7TT exercise bike. Maximum heart rate was determined by using a heart rate monitor (PolarH10, Finland). It began with a warm-up without load that lasted 5 min; subsequently, every two minutes, the intensity of the exercise was increased by 25 W, trying to maintain a cadence of 60 rpm. The test concluded when the subject being tested was unable to maintain cadence. The maximum heart rate obtained was determined by analyzing the data and verifying what the maximum was during the execution of the test. The test was supervised by a cardiologist and was carried out in the exercise physiology laboratory of the Universidad Autónoma de Chile, Temuco campus.

### 2.5. Instruments

The immersive VR instrument used was the Oculus Meta Guest 2 (Meta Platforms, Inc., Merlo Park, CA, USA), along with two touch controls.

During each session, using an ActiGraph wGT3X-BT (AG; ActiGraph, Pensacola, FL, USA), which has high levels of validity and reliability [45,46], the number of minutes that the study subject executed each of the intensities of physical activity (light, moderate, and vigorous) was determined. Data were collected at 100 Hz, using cut-off points proposed by Arias-Palencia et al. [47].

## 3. Results

Table 1 shows the results of the study subject for the pre- and post-intervention anthropometric variables. When comparing, after the intervention there was a reduction in body circumference and skinfold values.

Figure 1 shows the intensities of physical activity divided into light and moderate-vigorous. It can be observed that in most sessions the moderate-vigorous intensity time exceeds the light intensity.

Table 2 shows the values obtained for the study variables before the execution of the exercise program, those obtained once it was completed, and the percentage of change between both moments. For physical fitness, the hand-grip strength and speed-agility components decreased, while the cardiorespiratory fitness component increased. In terms of body composition, fat decreased, while muscle had a minimal increase.

## 4. Discussion

IVR-based training demonstrated better outcomes of some components of fitness and adiposity in a Physical Education student. Among the improvements observed are the decrease in the speed-agility test (−4.3%), the percentage of fat (−2.6%), and the kilos of fat (−14%) as well as the increase in performance in the cardiorespiratory fitness test (20.3%), the maximum volume of oxygen (12%), and the percentage and kilograms of muscle mass (0.1%). A curious result was the decrease in manual grip strength (−2% and −7.6%).

The literature has demonstrated the potential of IVR games to increase energy expenditure, decrease sedentary behaviors, and provide benefits in terms of body composition and motor competence [8]. In the field of rehabilitation, several researchers have implemented virtual reality interventions in older adults and women with fibromyalgia, combining the characteristics of physical exercise and minimizing pain or fear of the practice [6,9,10]. The results of these interventions demonstrate effectiveness in different components of physical fitness; in addition, they present a high adherence to the intervention, a favorable situation since this is usually a major problem in exercise interventions [6,9,10]. In this sense, there is evidence that virtual reality offers positive experiences, which increase motivation and effort and generate greater commitment than traditional exercise methods [48]. On the other hand, the use of these devices has also impacted the healthy population, even university students, as a recent review points to results such as those described above, demonstrating that IVR games have a similar potential to traditional exercise and can become a new and varied exercise option for sedentary people [31].

In relation to body composition, reductions in fat mass were found (14%), which is explained by the relationship between the increase in cardiorespiratory fitness and the reduction in fat mass [2]. Muscle mass presented a minimal increase (0.1%), a situation that can be explained by the characteristics of the application used (FITxR), which focuses mostly on the work of cardiorespiratory fitness, not giving great emphasis to muscle work; this is likely because one of the best ways to increase muscle mass is work through specific bodybuilding exercises with extra weight, which is difficult to do through IVR.

The results of this study demonstrate improvements of 20% in cardiorespiratory fitness, which is higher than that presented in other studies with middle-aged adults, where improvements ranged from 14% to 17% [49,50,51]. These differences may be due to the virtual reality device used, in this case Kinect, which offers a non-immersive virtual reality experience, unlike the one offered by the device used in this research (Oculus Meta Quest 2). In this sense, although the studies that use non-immersive virtual reality are similar to those of the present study in terms of frequency and duration, the differences in the interventions offered in the studies lie in the level of involvement, the intensity of the exercise, and the sensation of presence that they offer to the subject in terms of the type of exercise experience, which could increase effectiveness [52,53].

Another outstanding result is the improvement in agility-speed, reducing the seconds of execution at the end of the intervention by 4.3%. Most studies evaluating this variable focus on children and young people, since in these populations adequate values of this ability are related to higher bone mineral density and accumulation of bone mass at later ages [54]. This study contributes to the exploration of the behavior of this variable in adults; thus, it is recommended to incorporate them in future studies.

An unexpected result was the reduction of manual grip strength in both hands (right −2% and left −7.6%). These results are opposite to those obtained in the school population, where there was an improvement in performance in the hand-grip test [55]. This difference may be explained by the characteristics in the application used (FITXr), since it does not have actions that focus specifically on the strength work of the upper limbs, specifically the hand grip force, as it is based on making combinations of blows to spherical objects. Because of this, it is suggested that this type of intervention be complemented with specific strength exercises, to increase the health benefits of the participants. In addition, future studies should explore in greater detail the behavior of this ability to determine exactly the impact caused by virtual reality, and in this way consider whether training through IVR should be complemented with other strategies, including strength work.

Like any study, this one has limitations. First, the amount of physical activity performed by the subject outside of the intervention was not controlled; however, the subject was asked not to increase his levels of physical activity during the intervention. In addition, the subject, due to his professional training (Pedagogy in Physical Education), has practical subjects, which can affect the results. Finally, other factors that influence the variables of the study were not measured, such as eating habits, hours of sleep, and stress typical of university life.

Despite these weaknesses, the present study has great strengths, which lie in its methodological design, since standardized, valid, and reliable tests were used to measure the study variables. In addition, we maintained in the case study methodological rigor in the development of the intervention and in the analysis of the information. However, although the results are promising, they should be analyzed with caution, as due to the nature of case studies, their results are inconclusive [56].

## 5. Conclusions

Training developed through immersive virtual reality has proven to be a tool that can promote improvements in physical fitness and body composition. The improvements observed in the reduction of fat tissue and the increase in cardiorespiratory fitness and speed-agility demonstrate the potential of immersive virtual reality in the field of physical exercise. More research is needed to validate the potential of this instrument as a means that contributes to the health of the population.

## Figures and Tables

**Figure 1 jfmk-10-00056-f001:**
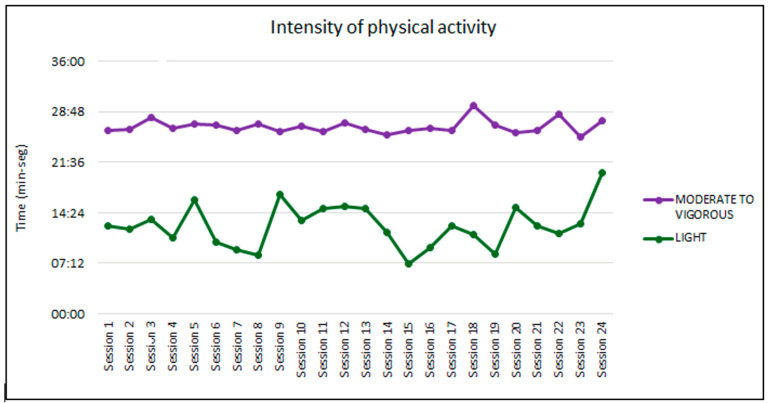
Intensity of physical activity for each session.

**Table 1 jfmk-10-00056-t001:** Characteristics of the study subject.

Anthropometric Variables	Pre-Intervention	Post-Intervention
Body mass (kg)	72.4	70.4
height (cm)	174	174
Sitting height (cm)	89.5	89.5
**Breadths**		
Biacromiale (cm)	41.2	41.2
Transverse chest (cm)	28.5	28.5
Antero-post chest (cm)	19.9	19.9
Bi-cristale (cm)	28.7	28.7
Humerus (cm)	7.4	7.4
Femur (cm)	10.7	10.7
**Girths**		
Head (cm)	57.2	57.2
Arm, relaxed (cm)	31.4	30.3
Arm flexed (cm)	33.9	33.5
Forearm (cm)	27.1	27.2
Chest (cm)	95.1	93.5
Waist (cm)	78.8	77.5
Hip (cm)	94.5	94.2
Upper thigh (cm)	55.5	55.3
Mid-thigh (cm)	52.1	51.3
Calf (cm)	36	35.2
**Skinfolds**		
Triceps (mm)	8.8	7.8
Subscapulare (mm)	10.8	10
Supraspinale (mm)	7.8	7
Abdominale (mm)	16.6	13
Thigh (mm)	13	12.4
Calf (mm)	6.8	7.4

**Table 2 jfmk-10-00056-t002:** Pre- and post-intervention differences of the study subject.

	Pre	Post	% Change
**Physical fitness**			
Right hand grip strength (kg)	44.2	43.3	−2
Left hand grip strength (kg)	40.7	37.6	−7.6
Speed-agility (s)	10.58	10.12	−4.3
Cardiorespiratory fitness (min)	9.35	11.25	20.3
VO_2_ (mL/kg/min)	47.6	53.6	12%
**Body Composition**			
Fat (%)	23.73	23.12	−2.6
Fat (kg)	19.45	16.73	−14
Muscle (%)	46.72	46.77	0.1
Muscle (kg)	33.82	33.85	0.1

VO_2_: Maximum oxygen volume.

## Data Availability

The data presented in this study are available on request from the corresponding author.

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
