# Peer review of "Impact on Body Composition and Physical Fitness of an Exercise Program Based on Immersive Virtual Reality: A Case Report"

_jfmk, 2025, doi:10.3390/jfmk10010056_

Round 1

Reviewer 1 Report

Comments and Suggestions for Authors

This manuscript investigates the effect of immersive virtual reality (IVR)-based exercise intervention on body composition and physical fitness through a case study. Considering the recent demand for smart technology in physical exercise programs, this case study explores a new and interesting trend. However, there are significant concerns regarding the manuscript's structure and the completeness of the data. Due to these critical issues, I recommend rejecting this manuscript for publication in JFMK. The authors should address these concerns through substantial revisions to improve the manuscript.

1.     In the abstract, please provide more details about the subject to enhance the informativeness of this case study.

2.     The first paragraph of the introduction lacks clarity and is overly long. The authors should consider restructuring it to provide a clearer context.

3.     In the introduction (Line 56), what is the definition of "young adult" in this study? Please specify.

4.     The sentence in Lines 58–61 seems to refer to the effects of COVID-19 on young adults. If this is the case, the authors should elaborate on these effects and include appropriate references.

5.     References should be added to support the statement in Lines 62–65.

6.     The third paragraph of the introduction includes a sentence that is too long, reducing readability. Please revise for clarity and conciseness.

7.     Physical fitness is the primary outcome measure in this study. The authors should provide more detailed descriptions of the physical fitness tests used, along with relevant references for the measurement methods.

8.     There are concerns regarding Table 2. This does not represent results of the intervention. It may be more appropriate to include it as part of the methods section related to the intervention sessions. Additionally, there appears to be no significant increase in physical activity duration over the sessions.

9.     In the section of results, the authors described about Figure 1. However, there is no figure included in the submitted manuscript.

10.  Please ensure consistent use of nomenclature and notation in Tables 1 and 3, particularly with commas and colons between numbers.

11.  In the third paragraph of the discussion (Lines 204–206), the authors claim that the IVR intervention is more effective in improving cardiovascular fitness compared to non-IVR interventions. If so, the authors should discuss the exercise components (e.g., intensity, duration, and frequency) of IVR versus non-IVR exercise programs.

12.  In the final part of the discussion, the authors should address the study's limitations, particularly the weak points of using a single-subject case study design.

Author Response

This manuscript investigates the effect of immersive virtual reality (IVR)-based exercise intervention on body composition and physical fitness through a case study. Considering the recent demand for smart technology in physical exercise programs, this case study explores a new and interesting trend. However, there are significant concerns regarding the manuscript's structure and the completeness of the data. Due to these critical issues, I recommend rejecting this manuscript for publication in JFMK. The authors should address these concerns through substantial revisions to improve the manuscript.

  1. In the abstract, please provide more details about the subject to enhance the informativeness of this case study.

A: Thank you for your comment. We have added information as suggested.

  1. The first paragraph of the introduction lacks clarity and is overly long. The authors should consider restructuring it to provide a clearer context.

A: Thank you for your comment. The first paragraph has been written.

  1. In the introduction (Line 56), what is the definition of "young adult" in this study? Please specify.

A: Thank you for your comment. It is incorporated into the text that defines the age between 18 and 25 years.

  1. The sentence in Lines 58–61 seems to refer to the effects of COVID-19 on young adults. If this is the case, the authors should elaborate on these effects and include appropriate references.

A: Thank you for your comment. Details were included and quotes were incorporated

  1. References should be added to support the statement in Lines 62–65.

A: A: Thank you for your comment. Quotes were incorporated.

  1. The third paragraph of the introduction includes a sentence that is too long, reducing readability. Please revise for clarity and conciseness.

A: A: Thank you for your comment. Paragraph is rewritten.

  1. Physical fitness is the primary outcome measure in this study. The authors should provide more detailed descriptions of the physical fitness tests used, along with relevant references for the measurement methods.

A: Thank you for your comment. We've added information regarding what is recommended.

  1. There are concerns regarding Table 2. This does not represent results of the intervention. It may be more appropriate to include it as part of the methods section related to the intervention sessions. Additionally, there appears to be no significant increase in physical activity duration over the sessions.

A: Thank you for your comment. It was moved to the methodology chapter, informing that if you want to know the minutes by the intensity generated in each session developed, it can be consulted in supplementary table II.

  1. In the section of results, the authors described about Figure 1. However, there is no figure included in the submitted manuscript.

A: Thank you for your comment. We forgot to include it, we regret the mistake. Now it was incorporated into the manuscript.

  1. Please ensure consistent use of nomenclature and notation in Tables 1 and 3, particularly with commas and colons between numbers.

A: Thank you for your comment. We have fixed the bugs.

  1. In the third paragraph of the discussion (Lines 204–206), the authors claim that the IVR intervention is more effective in improving cardiovascular fitness compared to non-IVR interventions. If so, the authors should discuss the exercise components (e.g., intensity, duration, and frequency) of IVR versus non-IVR exercise programs.

A: Thank you for your comment. As suggested, we have added information in the discussion chapter.

  1. In the final part of the discussion, the authors should address the study's limitations, particularly the weak points of using a single-subject case study design.

A: Thank you for your comment. We've added recommended information.

Reviewer 2 Report

Comments and Suggestions for Authors

Basic reporting

Dear authors, the manuscript is generally well-written and easy to read; a slight spell-check is required. I have some concerns that must be address. 

Introduction

The literature on the subject is sufficiently well summarised. However, it could be useful to clarify some information about:

-          statements like "young adults belonging to Generation Z are characterized by higher rates of mental health diagnoses and poor habits" may be an overgeneralization. You should account for potential variability within this population.

-          How IVR specifically reduces barriers like "long working hours" or "lack of motivation" could be explained more in depth.

-          IVR is presented as a promising tool but the rationale for selecting it over other interventions (e.g., traditional fitness programs or other digital tools) is not clearly justified.

-          You suggest that the target population will find IVR more engaging and effective without considering potential challenges (e.g., motion sickness, access to VR equipment, or individual preferences).

Methods

-          You suggest that the exercise program will be effective due to its intensity (moderate to vigorous). However, you should discuss whether the subject's baseline fitness level or other individual factors (e.g., motivation, preferences) might influence outcomes.

-          It’s not clear if the subject practised any other type of physical activity or sport during the intervention period, or whether his only activity was the one practised during this period.

-          There is no mention of reliability or calibration of the instruments like the ActiGraph and tests (4x10 test, Course Navette).

-          Were tests conducted under the same conditions pre- and post-intervention?

-          Was a familiarization session performed for physical fitness tests?

-          How was the subject’s maximum heart rate determined? (e.g., formula-based estimation or actual testing).

Validity of the findings

-          Grip strength and speed/agility decreased, while cardiorespiratory fitness improved. These results suggest inconsistent outcomes of the intervention. How did you explain strength and agility declining despite regular physical activity?

-          Fat reduction (-14%) and VO2 increase (+12%) suggest significant physical improvements. However, muscle mass change (+0.1%) is negligible, which seems inconsistent with fat loss.

-          The reduction in body mass (-2 kg) and circumferences is attributed to the intervention, but the role of potential confounders (e.g., dietary habits, stress, or other activities) should be accounted.

-          Please, check for typos (i.e., Table I "heigth" should be "height").

-          MVPA values seems inconsistent. MVPA for Session 1 (13:33) is lower than the sum of Moderate (13:19) and Vigorous (0:52), which should equal 14:11.

-          Skinfold measurements are subject to operator error and variability. Was the same evaluator used pre- and post-intervention? Were inter-day reliability assessments conducted?

-          You highlight improvements in cardiorespiratory fitness, fat reduction, and speed/agility, while acknowledging a decrease in grip strength. However, you don’t address the possible implications of these conflicting results. (i.e. Could the decrease in grip strength indicate a limitation of the program?         Does this suggest the need for supplementary exercises to target neglected fitness areas?)

-          The claim that IVR can “replace traditional exercise” is very speculative. This claim requires stronger empirical support, particularly given the mixed results and the single-subject design.

-          You mention IVR's potential to improve adherence but there are no data for supports this claim. Including the participant's qualitative feedback or adherence data would strengthen this point.

-          AS previously said, you should address potential confounding factors like dietary intake, sleep patterns, or other activities that might have influenced the results.

-          The speed/agility improvement is emphasized without no mechanism’s hypothesis.

-          The negligible increase in muscle mass (0.1%) and reduction in fat are not fully explored.

Author Response

Basic reporting 

Dear authors, the manuscript is generally well-written and easy to read; a slight spell-check is required. I have some concerns that must be address.  

Introduction 

The literature on the subject is sufficiently well summarised. However, it could be useful to clarify some information about:

-          statements like "young adults belonging to Generation Z are characterized by higher rates of mental health diagnoses and poor habits" may be an overgeneralization. You should account for potential variability within this population.

A: There is evidence that this population has characteristics influenced by their habits and lifestyles. However, within this same there is variability, but studies tend to generalize. To avoid confusion, the wording of the idea is improved. 

-          How IVR specifically reduces barriers like "long working hours" or "lack of motivation" could be explained more in depth.

A: The IVR does not reduce the barriers of long working hours, but there has been an increase in adherence to physical activity programs, mainly due to its recreational component. Improved wording to clarify the idea. We will not talk about barriers, but about increasing physical activity.

-          IVR is presented as a promising tool but the rationale for selecting it over other interventions (e.g., traditional fitness programs or other digital tools) is not clearly justified.

A: Thank you for your comment. We've added information regarding what is recommended.

-          You suggest that the target population will find IVR more engaging and effective without considering potential challenges (e.g., motion sickness, access to VR equipment, or individual preferences).

A: Thank you for your comment. We've added information regarding what is recommended.

Methods

-          You suggest that the exercise program will be effective due to its intensity (moderate to vigorous). However, you should discuss whether the subject's baseline fitness level or other individual factors (e.g., motivation, preferences) might influence outcomes.

A: Thank you for your comment. We've added recommended information.

-          It’s not clear if the subject practised any other type of physical activity or sport during the intervention period, or whether his only activity was the one practised during this period.

A: Thank you for your comment. We have added more information that describes greater characteristics of the study subject.

-          There is no mention of reliability or calibration of the instruments like the ActiGraph and tests (4x10 test, Course Navette).

A: Thank you for your comment. We've added the recommended information.

-          Were tests conducted under the same conditions pre- and post-intervention?

A: Thank you for your comment. Information on this is included in the chapter on Methodology.

-          Was a familiarization session performed for physical fitness tests?

A: Thank you for your comment. In the methodology chapter, information was added.

-          How was the subject’s maximum heart rate determined? (e.g., formula-based estimation or actual testing).

A: Thank you for your comment. Information was added as requested.

Validity of the findings

-          Grip strength and speed/agility decreased, while cardiorespiratory fitness improved. These results suggest inconsistent outcomes of the intervention. How did you explain strength and agility declining despite regular physical activity?

A: Thank you for your comment. Added information in the discussion regarding what is recommended.

-          Fat reduction (-14%) and VO2 increase (+12%) suggest significant physical improvements. However, muscle mass change (+0.1%) is negligible, which seems inconsistent with fat loss.

A: Thank you for your comment. Added information in the discussion regarding what is recommended.

-          The reduction in body mass (-2 kg) and circumferences is attributed to the intervention, but the role of potential confounders (e.g., dietary habits, stress, or other activities) should be accounted.

A: Thank you for your comment. Added information in the discussion regarding what is recommended.

-          Please, check for typos (i.e., Table I "heigth" should be "height").

A: Thank you for your comment. We've fixed that.

-          MVPA values seems inconsistent. MVPA for Session 1 (13:33) is lower than the sum of Moderate (13:19) and Vigorous (0:52), which should equal 14:11.

A: Thank you for your comment. We've fixed that.

-          Skinfold measurements are subject to operator error and variability. Was the same evaluator used pre- and post-intervention? Were inter-day reliability assessments conducted?

A: Thank you for your comment. In the methodology chapter, information on the characteristics of the evaluator was added.

-          You highlight improvements in cardiorespiratory fitness, fat reduction, and speed/agility, while acknowledging a decrease in grip strength. However, you don’t address the possible implications of these conflicting results. (i.e. Could the decrease in grip strength indicate a limitation of the program?         Does this suggest the need for supplementary exercises to target neglected fitness areas?)

A: Thank you for your comment. Added information in the discussion regarding what is recommended.

-          The claim that IVR can “replace traditional exercise” is very speculative. This claim requires stronger empirical support, particularly given the mixed results and the single-subject design.

A: Thank you for your comment. We have improved the wording considering the suggestion received.

-          You mention IVR's potential to improve adherence but there are no data for supports this claim. Including the participant's qualitative feedback or adherence data would strengthen this point.

A: Thank you for your comment. Added information regarding what is recommended.

-          AS previously said, you should address potential confounding factors like dietary intake, sleep patterns, or other activities that might have influenced the results.

A: Thank you for your comment. Information was added as recommended.

-          The speed/agility improvement is emphasized without no mechanism’s hypothesis.

Thank you for your comment. Added information.

-          The negligible increase in muscle mass (0.1%) and reduction in fat are not fully explored.

A: Thank you for your comment. Added information in the discussion chapter.

Round 2

Reviewer 1 Report

Comments and Suggestions for Authors

Thank you for the revised manuscript. The authors have sincerely addressed the comments and incorporated the suggested revisions. I am satisfied with their responses to the concerns raised in my initial review. The manuscript has been significantly improved overall. 

I recommend that the revised paper be accepted for publication.

Reviewer 2 Report

Comments and Suggestions for Authors

the authors adressed al my concerns. I have no further suggestion.